**Data Availability Statement:** All relevant data are within the manuscript.

# Factors influencing community acceptability of mass drug administration for the elimination of onchocerciasis in the Asante Akim South Municipal, Ghana

**Emmanuel Kumah** [1]*, **Patrick Owusu**[2], **Godfred Otchere**[3], **Samuel E. Ankomah**[4], **Adam Fusheini**[4,5], **Collins Kokuro**[6], **Frederick Mensah-Acheampong**[7], **Joseph Amankwah Atta**[8], **Samuel Kofi Agyei**[9]

1 Department of Health Administration and Education, Faculty of Science Education, University of Education, Winneba, Ghana, 2 Department of Public Health, Faculty of Health Sciences, Catholic University College of Ghana, Sunyani, Ghana, 3 Faculty of Humanities, Center for Medicine and Society, University of Freiburg, Freiburg, Germany, 4 Department of Preventive and Social Medicine, Dunedin School of Medicine, University of Otago, Dunedin, New Zealand, 5 Center for Health Literacy and Rural Health Promotion, Accra, Ghana, 6 Department of Medicine, School of Medicine and Dentistry, Kwame Nkrumah University of Science and Technology, Kumasi, Ghana, 7 Department of Administration, Cape Coast Teaching Hospital, Cape Coast, Ghana, 8 School of Public Health, Kwame Nkrumah University of Science and Technology, Kumasi, Ghana, 9 Department of Physician Assistantship, Faculty of Health and Medical Sciences, Presbyterian University College of Ghana, Asante Akyem Campus, Agogo, Ghana

* emmanuelkumah@uew.edu.gh, ababiohemmanuel@gmail.com

## Abstract

### Introduction

Onchocerciasis is one of the eleven neglected tropical diseases (NTDs) recently targeted by the World Health Organization (WHO) for elimination. Mass drug administration (MDA) of ivermectin has become the main intervention for reducing the burden of onchocerciasis and controlling its transmission. However, despite the considerable gains in the fight against onchocerciasis in Ghana, the infection remains endemic in some communities. This study aimed to ascertain community members' acceptability levels and factors associated with ivermectin MDA for the elimination of onchocerciasis in the Asante Akim South Municipal in the Ashanti Region of Ghana.

### Methods

A cross-sectional, questionnaire-based study was conducted in six communities in the Asante Akim South Municipal from 7th May to 9th July, 2021. The study population comprised all persons aged 18 years and above who had lived in the study communities for more than three months prior to the study. The main outcome variable was the acceptability of ivermectin MDA by the community members. This was measured using a composite acceptability score adapted from the Intervention Rating Profile tool. The explanatory variables were the respondents' socio-demographic characteristics, self-rated knowledge of onchocerciasis, perceived side effects of ivermectin, and self-reported level of education received on MDA activities.

**Funding:** The author(s) received no specific funding for this work.

**Competing interests:** The authors declare that they have no competing interests

## Results

Out of 450 community members included in the study, 50.4% were male while 49.6% were female. The respondents' mean age was 39.57±10.64 years. The mean acceptability score for ivermectin MDA was 20.52±2.91 (range, 9–36). Acceptability of ivermectin MDA was positively associated with gender, educational status, employment status, self-rated knowledge of onchocerciasis, and level of education received on MDA; and negatively associated with perceived side effect of ivermectin.

## Conclusion

This study provides valuable information to inform policy decisions on planning and implementing MDA programs for the elimination of onchocerciasis in the study area and the country as a whole.

### Author summary

Mass drug administration (MDA) using ivermectin is considered the main intervention for controlling the transmission of onchocerciasis. The program involves administering the drug to an entire target population irrespective of infection status or symptom. In most instances, administration of ivermectin is done annually in areas with low transmission and biannually in high transmission settings. However, when a proportion of the population fails to participate in an MDA program, a potential reservoir for the parasite is left untreated, opening the door to recrudescence of microfilaremia and reducing the probability of successful elimination of the disease. Thus, community participation is considered vital in addressing the gaps in MDA coverage and uptake. There is a link between participation and acceptability of a health intervention. For instance, once communities deem MDA to be acceptable, they are more likely to be motivated to participate in the intervention. Measures of acceptability provide a clearer understanding of factors that might be associated with compliance with a heath intervention. This study, therefore, assesses factors influencing community acceptability of MDA for the elimination of onchocerciasis. The findings provide useful information that could inform policy decisions on strategies to address MDA implementation gaps.

## Introduction

Onchocerciasis is one of the eleven neglected tropical diseases (NTDs) recently targeted by the World Health Organization (WHO) for elimination [1]. According to WHO, around 120 million people worldwide are at risk of onchocerciasis, among which 96% are in Africa [2]. Also, a report by the Global Burden of Disease Study estimated in 2017 that there were 20.9 million prevalent *Onchocerca* volvulus infections worldwide with 14.6 million people infected with skin disease and 1.15 million people infected with vision loss [3].

Commonly known as river blindness, onchocerciasis is a vector-borne disease caused by the filarial nematode known as *Onchocerca volvulus* and transmitted by a black fly of the genus *Simulium* [4,5]. Medically, onchocerciasis causes severe pruritus, nodules, dermatitis, depigmentation and, in the final stage, blindness [6]. On the socio-economic front, it leads to abandonment of work, depopulation of fertile lands, and many other consequences, such as

rejection and stigmatization, especially when the afflicted becomes blind [7]. The disease mostly affects rural communities and is a major cause of blindness and skin disease in endemic areas with serious socioeconomic consequences [3, 5].

In Ghana, onchocerciasis remains a significant public health problem. The country has been endemic to the condition since the early 1960s, and has been involved in its control since the inception of a WHO-led program known as the Onchocerciasis Control Program (OCP) which operated between 1972 and 2002. Control strategies have included removal of nodules (nodulectomy), vector control and mass drug administration (MDA) using ivermectin [8].

MDA of ivermectin has become the main intervention for reducing the burden of onchocerciasis and controlling its transmission [9, 10]. The program involves the administration of ivermectin to the entire target population irrespective of infection status or symptom [11]. In Ghana, MDA activities are organized annually in areas considered hypoendemic (microfilaria prevalence ≥30%≤ 35%) and mesoendemic (microfilaria prevalence >35% <60%), and biannually in hyperendemic settings (microfilaria prevalence ≥ 60%) [12]. Prior to the MDA program, community drug distributors (CDD) are selected from the communities involved and trained on how to distribute the drugs. The communities are then informed about the program by the local health authorities through social gatherings (such as community durbars), health outreach programs, and the mass media, including local radio and television stations, and newspapers [12].

Despite the considerable gains in the fight against onchocerciasis, the infection remains endemic in some communities in Ghana. For instance, community prevalence of microfilaria ranges from 2.4% to 13.2%, while that for nodule ranges from 4.9% to 14.3% [12]. This suggests that the acceptability and drug uptake is low in these endemic communities. A study conducted among people living in the Ankobra community in the south-western part of Ghana found that 33.3% of the eligible population did not comply with the MDA program [13]. Recently, Osei et al. investigated the extent and predictors of ivermectin coverage and uptake in endemic communities within the Atwima Nwabiagya North District in the Ashanti Region of Ghana and observed that 63.9% of the respondents did not receive the drug during the 2019 MDA program and more than half were not aware of the program [14].

Community participation is considered important in addressing the gaps in MDA coverage and uptake [15]. When a proportion of the population fails to participate in MDA with ivermectin, a potential reservoir for the parasite is left untreated, opening the door to recrudescence of microfilaremia and reducing the probability of successful elimination of the disease [2].

Treatment acceptability, as defined by Sekhon et al. is *the extent to which people consider an intervention to be appropriate based on their cognitive or emotional response to that intervention* [16]. Acceptability incorporates factors that go beyond coverage, the most commonly used measure of the success of MDA programs. Measures of acceptability provide a more nuanced understanding of factors that might be associated with compliance with a heath intervention [17]. Once communities deem MDA to be acceptable, they are more likely to be motivated to participate in the intervention [18]. Previous studies have reported several factors associated with the acceptability of a health intervention. These include the characteristics of the treatment used in the intervention, such as the treatment formulation, application, perceived benefit and the process used to deliver the treatment [19]. Other factors influencing acceptability include whether participants understand and feel capable of performing the behaviors required by the intervention; and socio-demographic characteristics of the target population, comprising age, gender, geographic location, employment status and educational level [20, 21].

The present study builds on these previous studies by ascertaining community members' acceptability levels and factors associated with ivermectin MDA for the elimination of

onchocerciasis in the Asante Akim South Municipal in the Ashanti Region of Ghana. Findings from the study provide useful baseline information that could inform the design of MDA strategies to ensure that implementation gaps are addressed in the country.

## Materials and methods

### Ethics statement and approvals

Ethical clearance was obtained from the Ethics Review Committee of the Ghana Health Service Research and Development Division, Accra (Protocol number GH-ERC 025/06/21) and Graduate and Research Department of Catholic University College of Ghana. In addition, the Municipal Health Director of Asante Akim South was formally written to for permission. We verbally informed the community leaders about the study process and possible outcomes and ensured an understanding was reached before commencing the study. All information captured was treated confidentially.

### Study design and setting

A cross-sectional, questionnaire-based study was conducted in six communities in the Asante Akim South Municipal in the Ashanti Region of Ghana from 7th May to 9th July, 2021. The municipal was selected for this study because of its endemicity for onchocerciasis, and due to reports indicating that previous MDA programs have recorded a geographical coverage below 80% [22]. Also, there have been reported instances of people refusing to accept and ingest ivermectin during MDA activities in the municipality [22]. MDA activities have been in place in the municipality since 1995 [22].

Asante Akim South is situated in the eastern part of the Ashanti Region [23]. It covers a total surface area of about 1,153.3 square kilometers, which forms about 5% of the total area of the Ashanti Region. It shares boundaries with the Asante Akim Central Municipal in the North, Asante Akim North Municipal in the Northwest and the Bosome-Freho District in the Southwest, all in Ashanti Region. With a population growth rate of 2.7%, the municipal has an estimated population of 148,780, representing 2.5% of the region's total population. About 83% of the population is rural, with the majority being male (51.3%). The municipal has been divided into six health sub-municipals to ensure effective and efficient health services delivery. These sub-municipals are: Banka, Bompata, Juaso, Komeso, Obogu and Ofoase [23].

### Study population

The target population for the study comprised all persons aged 18 years and above who had lived in the study communities for more than three months prior to the study. Community members below the age of 18 years and persons having mental problems were excluded from the study.

### Study variables

The main outcome variable for the study was the acceptability of ivermectin MDA by the community members. This was measured using a composite acceptability score (Table 1) adapted from the Intervention Rating Profile (IRP) tool [20]. The IRP tool consists of nine acceptability indicators scored across a four-point scale (i.e., disagree a lot = 1, disagree = 2, agree = 3, and agree a lot = 4) [17, 24]. The possible range of acceptability scores range from 9 to 36, with 22.5 being the threshold of acceptability [24].

The explanatory variables were mainly the respondents' socio-demographic characteristics, including age, gender, marital status, employment status, educational level, accommodation

**Table 1. The nine acceptability indicators combined to form the acceptability score.**

| No. | Indicator |
|---|---|
| 1 | This drug works against onchocerciasis or river blindness |
| 2 | This drug reduces the burden of itching |
| 3 | This drug works against the occurrence and severity of skin symptoms |
| 4 | I would take this treatment again |
| 5 | I would recommend this treatment to my relatives |
| 6 | I would be willing to change my family's routine so that we take the treatment again |
| 7 | The benefits of this treatment outweigh its side effects |
| 8 | This treatment is a good way to help our health problems here |
| 9 | Overall, mass drug administration of ivermectin will help my community |

Adapted from Carter [20]

status, religion, and duration of stay in the community. The other explanatory variables were self-rated knowledge of onchocerciasis as a disease, perceived side effects of ivermectin, and self-reported level of education received on MDA activities in the community. Age was grouped into <25 years, 26–40 years, 41–50 years, and >50 years. Marital status was defined as single, cohabiting, married, divorced or widowed. Employment status was categorized into unemployed and employed. Educational level was classified into no formal education, basic (primary and junior high school), secondary (completion of senior high school) and tertiary (attainment of post-secondary education). Accommodation status was grouped into living in a rented house, living in a family house, and living in a personal house. Religion was classified into Christian, Islam and Traditional. Duration of stay in the community was grouped into <10 years, 10–20 years and >20 years. Self-rated knowledge of onchocerciasis was categorized into no knowledge, little knowledge, average knowledge, good knowledge, and very good knowledge. Perceived side effects of ivermectin was classified into yes (meaning ingesting ivermectin has some side effects), maybe, and no (indicating no side effect). Finally, self-reported level of education received on MDA activities was grouped into no education at all, little education, average education and adequate education.

## Sample size estimation and sampling

A total of 450 community members were included in the study. This was determined using the Taro Yamane's approach [25]:

$$n = \frac{N}{1 + N(e)^2}$$

Where n = the expected sample size, N = finite population out of which the sample size is drawn (N = 148,780), and $e$ = level of precision (e = 0.05).

Substituting the above figures:

$$n = \frac{148780}{1 + 148780(0.05)^2}$$

$$n = \frac{148780}{372.95} = 399$$

The final 450 sample size was arrived at by adding a non-response rate of 10% and 11 more participants to ensure equal representation of the selected communities that participated in the study.

The selection of respondents involved a two-stage sampling process. First, stratified and simple random sampling techniques were employed to select six communities from the six sub-municipalities within the Asante Akim South Municipal. Names of the communities within each of the six sub-municipals were written on pieces of paper of the same size. These papers were folded, placed in six separate bowls, and vigorously shaken. A passer-by was asked to pick one of the papers from the first bowl and the name of the community picked was written down. The process was repeated until the six communities were selected. Consecutive sampling method was then used to select 75 eligible community members from each of the six selected communities. Consecutive sampling method is a sampling technique in which every subject meeting the criteria of inclusion is selected until the required sample size is achieved [26]. The method was chosen for this study because it allowed us to conveniently recruit eligible participants consecutively into the study until the desired sample size was attained.

## Study instrument and data collection

A 20-item questionnaire was developed and administered to collect data for the study. The questionnaire centered on the demographic characteristics of the respondents, the nine acceptability indicators (Table 1), and the three self-reported indicators about knowledge of onchocerciasis, perceived side effect of ivermectin, and level of education received on MDA activities. The questionnaire was pilot tested in one community in the study area. Necessary modifications were made before the actual data collection.

The administration of the questionnaire was done face-to-face, using a structured interview approach. Questions were read out and, in some instance, translated into the main local dialect (Twi) for those who could not speak English to understand and choose appropriate responses. Four research assistants were trained to assist in the data collection, which lasted approximately two months (from 7[th] May to 9[th] July, 2021). Oral informed consent was obtained from each participant before administering the questionnaire. The World Health Organization and the government of Ghana's COVID-19 preventive measures were strictly adhered to during the data collection period.

## Data processing and analysis

Data collected from the questionnaire administration was checked on the field by two members (EK and PO) of the research team, who supervised the data collection, to ensure completeness. This was done by going through each filled questionnaire to ensure all sections had been duly completed. The data was checked again by a third member (GO) of the research team and then coded. Data analysis involved three stages. First, descriptive statistics were computed on all of the selected variables. Second, a composite acceptability score was computed by linearly transforming the scores of the nine acceptability indicators to a 0–100 possible, and then averaging the items within the composite. Internal consistency reliability of the items measuring the composite variable was tested using Cronbach's alpha. The alpha value was 0.89, thus indicating a satisfactory result. Finally, using the composite acceptability score as the dependent variable, we run a multiple linear regression model, introducing the socio-demographic characteristics, self-rated knowledge of onchocerciasis, perceived side effects of ivermectin, and self-reported level of education received on MDA activities as predictors of community acceptability of ivermectin MDA. We examined normal probability plots of residuals and scatter diagrams of residuals versus predicted residuals to test the assumptions of

multiple regression analysis: normality, linearity and homoscedasticity. No violations were detected. A p-value of 0.05 was considered statistically significant. All analyses were done using STATA statistical software package version 16.0 (StataCorp. 4905 Lakeway Drive Station, Texas 77,845, USA).

## Results

### Sample population characteristics

Out of the 450 community members included in the study, 50.4% were male while 49.6% were female. The respondents' mean age was 39.57±10.64 years, with the majority falling within the age range of 26–40 years (38.4%). The majority indicated that they were married (53.9%), had basic level of education (56.8%), were employed (81.1%), and that they had lived in their community between 10 and 20 years (56.3%). Table 2 provides a detailed description of the respondents' socio-demographic characteristics.

Table 3 presents the respondents' self-reported knowledge of onchocerciasis, perceived side effects of ivermectin, and level of education received on MDA activities. Most of them rated their knowledge of onchocerciasis as little (26.1%) and average (31.6%). The majority (51.4%) responded in the affirmative that ivermectin has side effects. Regarding the level of education received on MDA activities, more than half (55.4%) reported that they had received little education, with only 12.5% indicating adequate knowledge of MDA activities in the community.

### Acceptability of mass drug administration of ivermectin for onchocerciasis elimination

The mean acceptability score for the nine items was 20.52±2.91 (range, 9–36). On a 1–4 scale, three indictors: "I would take this treatment again", "I would be willing to change my family's routine so that we take the treatment again", and "The benefits of this treatment outweigh its side effects" had acceptability scores below average (i.e., <2) (Table 4). Acceptability scores of the remaining six indicators were above average. The indicator "Overall, mass drug administration of ivermectin will help my community" had the highest acceptability score (3.18). This was followed by the indicator "This treatment is a good way to help our health problems here" (2.99). The indicator "I would take this treatment again" had the lowest acceptability score (1.61).

Table 5 shows standardized regression coefficients (β) and p-values of the independent variables. Explained variance for the regression model was 62.5%. Age, marital status, accommodation status, religious affiliation and period of stay in the community were not significantly associated with community acceptability of MDA of ivermectin. In total, six variables were statistically significant. Acceptability of MDA of ivermectin was positively associated with gender, educational status, employment status, self-rated knowledge of onchocerciasis, and level of education received on MDA; and negatively associated with perceived side effects of ivermectin. Female respondents tended to be more positive towards accepting MDA of ivermectin compared with their male counterparts (β = 1.72, p = 0.014). Also, respondents with higher levels of education tended to be more receptive to MDA of ivermectin compared to those with no formal education (β = 2.18, p = 0.021; β = 2.55, p = 0.032 and β = 3.12, p = 0.012 for basic, secondary and tertiary levels of education respectively). Furthermore, respondents who reported being employed tended to be more receptive to ivermectin MDA compared with the unemployed (β = 1.84, p = 0.042). Moreover, we observed that the higher the respondents rated their knowledge of onchocerciasis and the level of education they had received on MDA activities, the more positive they were towards the acceptability of MDA of ivermectin. Finally, respondents who perceived ivermectin to have side effects when ingested tended to be more

**Table 2. Characteristics of the study respondents.**

| Characteristic | | Frequency | Percentage |
|---|---|---|---|
| Age | | | |
| | < 25 years | 101 | 22.4 |
| | 26–40 years | 173 | 38.4 |
| | 41–50 years | 135 | 30.1 |
| | >50 years | 41 | 9.1 |
| Gender | | | |
| | Male | 227 | 49.6 |
| | Female | 223 | 50.4 |
| Marital Status | | | |
| | Single | 137 | 30.5 |
| | Cohabiting | 30 | 6.2 |
| | Married | 242 | 53.9 |
| | Divorced | 24 | 5.5 |
| | Widowed | 17 | 3.8 |
| Educational Status | | | |
| | No formal education | 81 | 17.9 |
| | Basic | 255 | 56.8 |
| | Secondary | 83 | 18.4 |
| | Tertiary | 31 | 6.9 |
| Employment Status | | | |
| | Employed | 85 | 18.9 |
| | Unemployed | 365 | 81.1 |
| Accommodation Status | | | |
| | Renting | 126 | 27.9 |
| | Family House | 218 | 48.4 |
| | Personal House | 106 | 23.6 |
| Religion | | | |
| | Christian | 321 | 71.3 |
| | Muslim | 118 | 26.3 |
| | Traditionalist | 11 | 2.4 |
| Period of stay in the community | | | |
| | <10 years | 64 | 14.1 |
| | 10–20 years | 253 | 56.3 |
| | >20 years | 133 | 29.6 |

negative towards the acceptability of MDA compared with those who indicated that ivermectin had no side effects (β = -10.48, p = 0.001).

## Discussion

One of the key principles underpinning the operations of MDA is ensuring that there is community ownership of the program [27]. The purpose of this study was to determine community members' acceptability of ivermectin MDA and its associated factors in the Asante Akim South Municipal in the Ashanti Region of Ghana. Using a mean acceptability score as the primary outcome of interest, a multiple linear regression model was fitted for ten independent variables. Mean acceptability score was 20.52±2.91 (range, 9–36). Acceptability of MDA of ivermectin was significantly associated with six variables: gender, educational status,

**Table 3. Knowledge, perceived side effect, and level of education received on ivermectin mass drug administration.**

| Indicator | | Frequency | Percentage |
|---|---|---|---|
| Knowledge of onchocerciasis: | | | |
| | No knowledge | 92 | 20.5 |
| | Little knowledge | 117 | 26.1 |
| | Average knowledge | 143 | 31.6 |
| | Good knowledge | 69 | 15.3 |
| | Very good knowledge | 29 | 6.4 |
| Perceived side effect of ivermectin: | | | |
| | No | 101 | 22.5 |
| | Maybe | 118 | 26.1 |
| | Yes | 231 | 51.4 |
| Education received on MDA: | | | |
| | No education at all | 53 | 11.8 |
| | Little education | 249 | 55.4 |
| | Average education | 92 | 20.3 |
| | Adequate education | 56 | 12.5 |

MDA = mass drug administration

employment status, self-rated knowledge of onchocerciasis, level of education received on MDA, and perceived side effects of ivermectin.

Acceptability is an important factor in determining compliance (i.e., actual users) with treatment interventions. High acceptability scores are associated with high compliance with MDA programs [18]. However, the mean acceptability score of 20.52 found in the present study is below the 22.5 threshold of acceptability of a health intervention [24]. This implies ivermectin MDA has low acceptability, and thus low compliance, in the studied communities. This observation supports an earlier study in the Central Region of Ghana which found a decline in mean compliance of ivermectin mass treatment from 39% to 26% within 7 years (2006–2013) although geographical converge (the eligible people who received the drug) increased to about 97% [28]. In a related study, Dicko et al. reported a geographical converge of 73.5%, but with only 66.6% compliance [13]. The observations imply that achieving high geographical converge alone, although necessary, may not translate into high acceptability and high compliance with MDA programs.

**Table 4. Mean acceptability scores of ivermectin mass drug administration (N = 450).**

| Indicator | Mean score (1–4) |
|---|---|
| This drug works against onchocerciasis or river blindness | 2.48 |
| This drug reduces the burden of itching | 2.32 |
| This drug works against the occurrence and severity of skin symptoms | 2.37 |
| I would take this treatment again | 1.61 |
| I would recommend this treatment to my relatives | 2.12 |
| I would be willing to change my family's routine so that we take the treatment again | 1.67 |
| The benefits of this treatment outweigh its side effects | 1.78 |
| This treatment is a good way to help our health problems here | 2.99 |
| Overall, mass drug administration of ivermectin will help my community | 3.18 |

Composite mean acceptability score = 20.52±2.91 (Range, 9–36)

**Table 5. Predictors of community acceptability of mass drug administration (n = 450).**

| Variable | | Standardized regression coefficient (range) | P-value |
|---|---|:---:|:---:|
| Age | | | |
| | < 25 years | Ref. | |
| | 26–40 years | 1.18(0.54–1.92) | 0.138 |
| | 41–50 years | 1.55(0.92–2.44) | 0.357 |
| | >50 years | 1.88(1.48–2.53) | 0.286 |
| Gender | | | |
| | Male | Ref. | |
| | Female | 1.7(1.24–2.32) | 0.014 |
| Marital Status | | | |
| | Single/separated | Ref. | |
| | Living with spouse | 1.27(0.91–1.30) | 0.422 |
| Educational Status | | | |
| | No formal education | Ref. | |
| | Basic | 2.18(1.54–2.92) | 0.021 |
| | Secondary | 2.55(1.92–3.43) | 0.032 |
| | Tertiary | 3.12(2.48–4.53) | 0.012 |
| Employment Status | | | |
| | Unemployed | Ref. | |
| | Employed | 1.84(1.42–2.54) | 0.042 |
| Accommodation Status | | | |
| | Renting | Ref. | |
| | Family/Personal House | 1.92(0.77–2.95) | 0.639 |
| Religion | | | |
| | Christian | Ref. | |
| | Non-Christian | 4.92(3.77–5.95) | 0.712 |
| Period of stay in the community | | | |
| | <10 years | Ref. | |
| | 10–20 years | 3.48(2.50–4.77) | 0.233 |
| | >20 years | -3.94(1.95–4.18) | 0.248 |
| Knowledge of onchocerciasis | | | |
| | No knowledge | Ref. | |
| | Little knowledge | 2.92(1.77–3.95) | 0.032 |
| | Average knowledge | 3.44(2.53–4.76) | 0.024 |
| | Good knowledge | 3.94(2.95–5.17) | 0.011 |
| | Very good knowledge | 4.04(3.15–5.97) | 0.012 |
| Perceived side effect of ivermectin | | | |
| | No | Ref. | |
| | Maybe | -9.24(7.13–11.55) | 0.004 |
| | Yes | -10.48(8.33–13.17) | 0.001 |
| Education received on MDA | | | |
| | No education at all | Ref. | |
| | Little education | 8.15(7.11–9.60) | 0.023 |
| | Average education | 9.65(8.71–11.20) | 0.002 |
| | Adequate education | 11.35(10.41–12.90) | 0.015 |

Multivariable linear regression model with a p-value $\leq$ 0.05 considered statistically significant

Knowledge of onchocerciasis is a useful measure towards the elimination of the disease. Studies have shown that the perception of community members on the risk of onchocerciasis may negatively affect participation in community-directed treatment with ivermectin [29]. It is therefore important that the knowledge of individuals living in onchocerciasis endemic areas is critically assessed. In the present study, a significant proportion of the respondents reported little (26.1%) to average (31.6%) knowledge of onchocerciasis. This is consistent with Agyemang et al's study which reported a decline in the proportion of community members with some knowledge of onchocerciasis in the Upper Denkyira East Municipal in the forest area of Ghana from 88% in 2006 to 64% in 2013 [28]. The finding is also consistent with other similar studies conducted in other African countries [4, 30]. These findings highlight the need for more aggressive advocacy, awareness, and sensitization programs in onchocerciasis endemic communities in Africa.

Related to the knowledge of onchocerciasis is the level of education received on MDA. More than half (55.4%) of the respondents reported that they had received little education, with only12.5% indicating adequate knowledge of MDA activities in their community. We also observed that the higher the respondents rated the level of education they had received on MDA activities, the more positive they were towards accepting MDA of ivermectin. These observations reinforce the need to intensify education and awareness creation about MDA activities in onchocerciasis endemic communities in the country. Evidence has shown that social mobilization and awareness campaigns to inform communities about the process before and during community-based interventions contribute to the success of MDA programs [31].

One interesting finding from this study is the inverse relationship between community acceptability of MDA and perceived side effects of ivermectin. Respondents who perceived ivermectin to have side effects when ingested tended to be more negative towards accepting the treatment intervention. This is not surprising because instances of alleged side effects after ingesting ivermectin have been documented. These, which have been reported mainly by community drug distributors (CDDs), include oedema, rashes, joint pains, reduction in body weight, dizziness, collapsing and the emergence of other diseases that led to the death of some people [28]. Though all these side effects are yet to be scientifically proven, it appears most community members believe them, hence their fear of accepting ivermectin MDA. In a recent study, Osei et al. observed that about half of the respondents who were given ivermectin in an MDA program refused to ingest the drug for fear of side effects [15]. Thus, to improve the acceptability of MDA, there is the need for rigorous community sensitization programs on the safety and possible side effects of ivermectin to allay the fears of community members.

## Limitations

There are several limitations to this study. First, the study does not represent the views of the entire population in Ashanti Region, but limited to only the study area. Hence, the findings may not be generalized. Recall bias is another problem as some members of the community had to recollect what happened during the last MDA program. Again, the cross-sectional nature of the study makes it limited in its ability to establish causality. Moreover, the single knowledge-rating item used in the study might not represent an adequate measure of the respondents' knowledge of onchocerciasis as a disease. Finally, we did not find enough previous studies on the topic to compare the mean acceptability score generated in this study. This limited our ability to do a detailed comparison of the findings with the existing literature. That notwithstanding, the study provides important background information for further studies on the topic, particularly within the Ghanaian context.

## Conclusions

This study has provided valuable information to inform policy decisions on planning and implementing MDA programs for the elimination of onchocerciasis in the study area and the country as a whole. We found that ivermectin MDA had low acceptability in the Asante Akim South Municipal. This low acceptability could be aassociated with the male gender, having low educational status, being unemployed, having little knowledge of onchocerciasis as a disease, receiving little education on MDA activities, and perceiving that ivermectin has side effects. These factors should be taken into consideration in the design and implementation of future MDA programs to improve community acceptability.

## Acknowledgments

We thank the community members within the Asante Akim South Municipal for participating in this study. We are also grateful to the research assistants who assisted us in collecting data for the study.

## Author Contributions

**Conceptualization:** Emmanuel Kumah, Patrick Owusu.

**Data curation:** Emmanuel Kumah, Patrick Owusu, Godfred Otchere, Samuel E. Ankomah, Adam Fusheini, Collins Kokuro, Frederick Mensah-Acheampong, Samuel Kofi Agyei.

**Formal analysis:** Emmanuel Kumah, Patrick Owusu, Godfred Otchere, Samuel E. Ankomah, Adam Fusheini, Collins Kokuro, Joseph Amankwah Atta.

**Investigation:** Emmanuel Kumah.

**Methodology:** Emmanuel Kumah, Samuel E. Ankomah, Adam Fusheini, Collins Kokuro, Frederick Mensah-Acheampong, Joseph Amankwah Atta, Samuel Kofi Agyei.

**Supervision:** Emmanuel Kumah.

**Validation:** Emmanuel Kumah.

**Writing – original draft:** Emmanuel Kumah, Patrick Owusu.

**Writing – review & editing:** Emmanuel Kumah, Samuel E. Ankomah, Adam Fusheini, Collins Kokuro, Frederick Mensah-Acheampong, Joseph Amankwah Atta, Samuel Kofi Agyei.

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

27706162

