## [Decision Letter · Decision Letter 0]

8 Dec 2022

Dear Dr Kumah,

Thank you very much for submitting your manuscript "Factors influencing community acceptability of mass drug administration for the elimination of onchocerciasis in the Asante Akim South Municipal, Ghana" for consideration at PLOS Neglected Tropical Diseases. As with all papers reviewed by the journal, your manuscript was reviewed by members of the editorial board and by several independent reviewers. In light of the reviews (below this email), we would like to invite the resubmission of a significantly-revised version that takes into account the reviewers' comments. 

We cannot make any decision about publication until we have seen the revised manuscript and your response to the reviewers' comments. Your revised manuscript is also likely to be sent to reviewers for further evaluation.

Sincerely,

Samuel Wanji

Academic Editor

Esther Schnettler

Section Editor

Reviewer's Responses to Questions

**Key Review Criteria Required for Acceptance?**

**Methods**

-Are the objectives of the study clearly articulated with a clear testable hypothesis stated?

-Is the study design appropriate to address the stated objectives?

-Is the population clearly described and appropriate for the hypothesis being tested?

-Is the sample size sufficient to ensure adequate power to address the hypothesis being tested?

-Were correct statistical analysis used to support conclusions?

-Are there concerns about ethical or regulatory requirements being met?

Reviewer #1: • The authors clearly stated the objective of the study

• Authors have obtained permission from the appropriate ethic regulatory bodies for the

study. However, there is no mention of whether permissions were sought from the chiefs

and opinion leaders from each of the communities where the study was conducted. I

suggest this information is included in the manuscript. Please let the ‘ethical

consideration’ statement appear at the first section under the methods and not to the end.

• The authors have not indicated what informed them to choose/select the study site. No

mention of the endemicity of the infection, ivermectin mass drug administration activities

if there is any, the coverage, compliance to ivermectin in previous or a recent mass drug

administration (MDA). Please I suggest the authors include these details in the study

design and setting to be able to address the stated objective.

• The authors clearly described the appropriate population suitable for the study. However,

the exclusion criterium in line 139 is not clear and should be stated with clarity. There is

no mention of seeking for consent from the participants before they were enrolled into the

study. I suggest that this information, either written or orally obtained should be included

in the study instrument and data collection section.

• The sample size was determined from sample size calculation and is sufficient to ensure

adequate power to address the hypothesis being tested.

• The authors have not mentioned the statistical test used to draw conclusion from the data.

I suggest this important information is included under the data processing and analysis.

Reviewer #2: THis is a very well written and presented manuscript on an important topic. 

The objectives and methods are clearly presented; there was not a clear hypothesis, but I do not view that as a deficiency. Description of study population, sample size and statistical analyses are good. I have no concerns about ethical approvals.

**Results**

-Does the analysis presented match the analysis plan?

-Are the results clearly and completely presented?

-Are the figures (Tables, Images) of sufficient quality for clarity?

Reviewer #1: • The results are not completely presented. I suggest that Table 5 is edited to include the

following:

- a column containing information on the P- values

- the third column that indicate the “B/ β”, should be written in full and write

“range” in bracket to let the readers understand the table without referring to the

manuscript

- in the results section of the manuscript, indicate the p values computed instead of

just writing ‘p’ is greater/lower than 0.05

- add to the footnote the statistical test used to calculate the p values

• Table 2 & 5 are overlapping. I suggest that if a table cannot fit into one page, the authors

should break that into sections. Example; Table 1, Table 1 (continued) maintaining the

title of the columns under each table.

Reviewer #2: Well described and tables are simple and clear.

**Conclusions**

-Are the conclusions supported by the data presented?

-Are the limitations of analysis clearly described?

-Do the authors discuss how these data can be helpful to advance our understanding of the topic under study?

-Is public health relevance addressed?

Reviewer #1: The authors provided conclusions which are supported by the data presented in the manuscript.

The limitations of the study were clearly described.

The benefit of the findings to the advancement of knowledge on the topic under study have been

discussed and public health relevance has also been touched on.

Reviewer #2: Conclusions and limitations well presented, and public health relevance is obvious (increased education and information distribution). I believe that the results will be beneficial to future MDA programs for oncocerciasis and other diseases.

**Editorial and Data Presentation Modifications?**

Reviewer #1: As commented under the result section, I suggest that tables in the manuscript that cannot fit into a page, should be broken down into sections to avoid overlapping. 

Table 5 should be edited to include the fields mentioned under the result section

Reviewer #2: I don't believe the word Ivermectin needs to be or should be capitalized

It appeaers that, likely through editing, multiple font sizes were used - please make font size consistent

Line 355: aassociated -> associated

**Summary and General Comments**

Reviewer #1: • The manuscript has not been edited and it contains different font sizes. I suggest authors

edit the manuscript to have a uniform font size.

• The in-text citation is incorrect. The journal requires a square bracket not parenthesis

• The authors should please provide citation to the statement made in line 83-86.

• I suggest the authors be specific on the kind of mass media being referred to at the end of

line 88 and 89.

• In line 92 and 93, the authors suggested low acceptability of MDA based on the

prevalence of a study stated in line 90 to 92. To make their suggestion more convincing, I

will recommend they mention the IVM coverage or compliance recorded in that study.

• Remove the letter “s” from the word “treatment’s” in line 111.

• Line 210 and 211. How were the data checks done, who did it? I recommend the authors

elaborate on this please.

• In line 215, the authors mentioned that the reliability of the items measuring the

composite variable was tested without mentioning how it was done. I suggest this should

be elaborated on.

• There is no numbering of the write-up after Table 2.

• What do the authors mean by using the word “significant” in the beginning of the second

sentence after Table 2? Was the statement based on statistically significant level. I

suggest the sentence should be rephrased.

• Add colon sign after the “Table 4” in the Table 4 heading (i.e., Table 4:).

• Replace “reeducation” in line 327 with “reduction”.

• No mention of authors contribution in the manuscript. I suggest authors include this

information after “Funding”

• Some of the bibliographies are incorrect. For example, “World Health Organization”

should either be written in full or abbreviated as “WHO” but not “World Health O”. I

recommend that the authors edit the bibliography to suite the journal’s style.

Reviewer #2: (No Response)

PLOS authors have the option to publish the peer review history of their article (what does this mean?). If published, this will include your full peer review and any attached files.

Reviewer #1: No

Reviewer #2: No
---

## [Decision Letter · Decision Letter 1]

10 Mar 2023

Dear Dr Kumah,

Thank you very much for submitting your manuscript "Factors influencing community acceptability of mass drug administration for the elimination of onchocerciasis in the Asante Akim South Municipal, Ghana" for consideration at PLOS Neglected Tropical Diseases. As with all papers reviewed by the journal, your manuscript was reviewed by members of the editorial board and by several independent reviewers. The reviewers appreciated the attention to an important topic. Based on the reviews, we are likely to accept this manuscript for publication, providing that you modify the manuscript according to the review recommendations. 

Sincerely,

Samuel Wanji

Academic Editor

Esther Schnettler

Section Editor

Reviewer's Responses to Questions

**Key Review Criteria Required for Acceptance?**

**Methods**

-Are the objectives of the study clearly articulated with a clear testable hypothesis stated?

-Is the study design appropriate to address the stated objectives?

-Is the population clearly described and appropriate for the hypothesis being tested?

-Is the sample size sufficient to ensure adequate power to address the hypothesis being tested?

-Were correct statistical analysis used to support conclusions?

-Are there concerns about ethical or regulatory requirements being met?

Reviewer #1: • Line 160-162: Kindly be specific on the type of inform consent sought from the community leaders (that is either verbal or written)

• Line 165-168: Please for how long has MDA activities/programs be in place in the selected municipality? Kindly mention this.

• Line 249: Kindly end the sentence with the word “period” to read “collection period”

Reviewer #2: This is a very clearly written and presented manuscript that quite adequately describes the survey that was conducted. The population under study is well described and descriptive statistics are appropriate. There do not appear to be any ethical concerns.

**Results**

-Does the analysis presented match the analysis plan?

-Are the results clearly and completely presented?

-Are the figures (Tables, Images) of sufficient quality for clarity?

Reviewer #1: Table 2: Kindly add “s” to the word “Characteristic” to read “Characteristics”

For the second Table 2 heading, kindly insert the "(continued)" after Table 2 to read “Table 2 (continued): Characteristics of the study respondents” but not at the end of the title. Please do same for the second Table 5 as well

Reviewer #2: Results and tables are very clearly presented.

**Conclusions**

-Are the conclusions supported by the data presented?

-Are the limitations of analysis clearly described?

-Do the authors discuss how these data can be helpful to advance our understanding of the topic under study?

-Is public health relevance addressed?

Reviewer #1: Please I do not have any comment

Reviewer #2: Conclusions and limitations are clearly presented, and the data coming from the survey leads to the (predicable) conclusion that the population, particularly unemployed males, need to somehow receive more education on the potential benefits of MDA and have their concerns about side effects mitigated. It is not clear how best to attain those goals, but that is not really part of this study.

**Editorial and Data Presentation Modifications?**

Reviewer #1: Please no comment

Reviewer #2: Very nicely presented - I do not have criticisms on presentation.

**Summary and General Comments**

Reviewer #1: Please no comment

Reviewer #2: A valuable survey that may provide hard data to stimulate more potent community outreach efforts for education on the benefits of MDA for onchocerciasis.

PLOS authors have the option to publish the peer review history of their article (what does this mean?). If published, this will include your full peer review and any attached files.

Reviewer #1: No

Reviewer #2: No

Figure Files:

Data Requirements:

Reproducibility:

References

---

## [Editor Report · Decision Letter 2]

17 Mar 2023

Dear Dr Kumah,

We are pleased to inform you that your manuscript 'Factors influencing community acceptability of mass drug administration for the elimination of onchocerciasis in the Asante Akim South Municipal, Ghana' has been provisionally accepted for publication in PLOS Neglected Tropical Diseases.

Best regards,

Samuel Wanji

Academic Editor

Esther Schnettler

Section Editor

---

## [Editor Report · Acceptance letter]

28 Mar 2023

Dear Dr Kumah,

We are delighted to inform you that your manuscript, "Factors influencing community acceptability of mass drug administration for the elimination of onchocerciasis in the Asante Akim South Municipal, Ghana," has been formally accepted for publication in PLOS Neglected Tropical Diseases.

Best regards,

Shaden Kamhawi

co-Editor-in-Chief

Paul Brindley

co-Editor-in-Chief
